# Transparent Polyimide/Organoclay Nanocomposite Films Containing Different Diamine Monomers

**DOI:** 10.3390/polym12010135

**Published:** 2020-01-06

**Authors:** Hyeon Il Shin, Jin-Hae Chang

**Affiliations:** Department of Polymer Science and Engineering, Kumoh National Institute of Technology, Gumi 39177, Korea; poweril55@naver.com

**Keywords:** transparent polyimide, nanocomposite, film, organoclay, thermo-mechanical properties, optical transparency

## Abstract

Poly (amic acid) s (PAAs) were synthesized using 4,4′-(hexafluoroisopropyl-idene) diphthalic anhydride (6FDA) and two types of diamines—bis(3-aminophenyl) sulfone (BAS) and bis(3-amino-4-hydroxyphenyl) sulfone (BAS-OH). Two series of transparent polyimide (PI) hybrid films were synthesized by solution intercalation polymerization and thermal imidization using various concentrations (from 0 to 1 wt%) of organically modified clay Cloisite 30B in PAA solution. The thermo-mechanical properties, morphology, and optical transparency of the hybrid films were observed. The transmission electronic microscopy (TEM) results showed that some of the clays were agglomerated, but most of them showed dispersed nanoscale clay. The effects of -OH groups on the properties of the two PI hybrids synthesized using BAS and BAS-OH monomers were compared. The BAS PI hybrids were superior to the BAS-OH PI hybrids in terms of thermal stability and optical transparency, but the BAS-OH PI hybrids exhibited higher glass transition temperatures (*T_g_*) and mechanical properties. Analysis of the thermal properties and tensile strength showed that the highest critical concentration of organoclay was 0.50 wt%.

## 1. Introduction

Polyimide (PI), which has been used for a long time, was mainly studied for military purposes when it was first developed [1,2]. It has also been studied in various applications in the aerospace industry and as electric materials. Moreover, PI thin films are easy to synthesize and do not require crosslinking agents for curing [3,4,5].

Recently, PI has been applied as an integrated material for semiconductor materials such as liquid crystal display (LCD) and plasma display panel (PDP) because of its low weight and its ability to improve electronic products. In addition, PI, which is lightweight and flexible, has been extensively studied for use in flexible plastic display substrates to overcome disadvantages such as heavy weight and tendency of shattering of glass substrates used in the field [6,7,8].

Although PI is a high-performance polymer material with excellent properties, including high thermal stability and good mechanical properties, chemical resistance, and electrical properties, it does not meet the basic requirements for display applications. Many synthesized PIs are insoluble and infusible and thus have poor processability. Therefore, efforts are now being made to improve these optical properties and the processability of PI. For example, PIs containing trifluoromethyl groups have been synthesized that show a high modulus, low thermal expansion coefficient, and good solubility in conventional organic solvents [9,10,11]. Another method is to use a copolyimide (Co-PI) using a specific monomer. A Co-PI typically possesses much lower molecular regularity than the corresponding homopolyimide [12,13]. This decreased regularity leads to fewer intermolecular interactions that, in turn, results in new characteristics, such as modified thermo-optical and gas permeation properties, and solubilities. Furthermore, the properties of Co-PIs can be adjusted by varying the ratio of the dianhydride and diamine comonomers.

PI is dark brown because electrons forming the double bonds in the benzene in the imide structure are transferred to the charge carriers of the π electrons generated by intermolecular bonding between the chain transfer complex (CT-complex) [14,15]. To reduce the CT-complex, a strong electron withdrawing group such as fluorine (F) or sulfone (-SO_2_-) is required that can effectively introduce a bending structure which can interfere with the interaction between the PI main chains, thus aiding in the fabrication of a PI film with high transparency. For example, a -CF_3_ group, which is a strong electron-withdrawing group, is often used as a substituent, or bent monomer structures are used to prevent CT complexes from being formed in linear structures [16,17].

It is also possible to improve the processability of PI films by introducing couplers with small polarity and free rotation in the main chain or by introducing bulky substituents to reduce the crystallinity and molecular packing density [18,19,20].

Among various layered inorganic materials used as nanofillers, clay has a layered structure with ionic bonds and Van der Waals forces acting between the layers. Clay has platelets with a thickness of 1 nm and a width of 100–1000 nm [21,22]. In general, layered inorganic materials are hydrophilic because the anions on the surface exist in a form that is stabilized with an alkali metal cation. Therefore, polymer materials including -OH groups are known to exhibit very good dispersibility and compatibility with clay [23,24]. Several methods are used to increase the physical properties of the blend, including hydrogen bonding using the hydroxyl (-OH) group between the polymer chain and the filler. For example, it is known that PVA can form a hydrogen bond between a polymer chain and a filler to ensure they are more tightly bonded, resulting in better physical properties than pure polymers [25,26].

In this study, the diamine monomers—bis(3-aminophenyl) sulfone (BAS) and bis(3-amino-4-hydroxyphenyl) (BAS-OH)—were allowed to react with 4,4′-(hexafluoroisopropyl (6FDA)) as a dianhydride to synthesize two PIs. PI hybrids were prepared from the synthesized PIs and Cloisite 30B as an organoclay. Cloisite 30B is hydrophilic because it has -OH groups. Furthermore, it is expected that the affinity of this clay will increase if it is blended with PI containing -OH groups. BAS-OH containing a hydroxyl group can enhance the reactivity, dispersibility, and compatibility between the synthesized PI and clay.

In this study, the thermo-mechanical properties, morphology, and optical transparency of the PI hybrids with various concentrations of organoclay (from 0 wt% to 1.00 wt%) were investigated, and their results were compared. We also described the effect of -OH groups in PI hybrids and compared the properties of the two PI hybrids synthesized using BAS and BAS-OH diamine monomers.

## 2. Experimental Details

### 2.1. Materials

Cloisite 30B was purchased from Southern Clay Product, Co. Most solvents used in this study were purchased from Daehan Chemical (Daegu, Korea). 6FDA, BAS, and BAS-OH were purchased from TCI (Tokyo, Japan) and Aldrich Chemical Co. (Yongin, Korea). DMAc was purified with 4Å molecular sieve and stored for water absorption treatment.

### 2.2. Preparation of PI Hybrid Films

Because the synthesis procedures of the two PAAs using BAS and BAS-OH diamine monomers are the same, only the method of fabricating BAS-OH PAA will be described here. BAS-OH (4.65 g, 1.85 × 10^−2^ mol) and DMAc (30 mL) were placed in a flask and stirred at 0 °C for 1 h while supplying nitrogen. A solution of 6FDA (7.38 g; 1.85 × 10^−2^ mol) in DMAc (40 mL) was added to the BAS-OH/DMAc solution. This solution was stirred at a moderate speed at 25 °C for 14 h to synthesize a PAA solution having a solid content of 15.5 wt%. The inherent viscosities of the synthesized PAAs of BAS and BAS-OH were 1.02 and 0.94, respectively. The total 15.5 wt% PAA solution produced by this method was 77.62 g. When 0.39 g of Cloisite 30 B was added to the solution, the total solution weight was 78.01 g.

Because the methods of synthesizing PI hybrids using different concentrations of organoclay were the same, the synthetic method of synthesizing PI/Cloisite 30 B (0.50 wt%) as a representative example will be described here. We first added 0.064 g of Cloisite 30 B, 13.0 g of PAA solution, and DMAc (20 mL) to a flask and vigorously stirred the mixture at room temperature for 3 h. The solution was also washed in an ultrasonic cleaner for 3 h, and the solvent was removed in a vacuum oven at 50 °C for 2 h. The obtained PAA film was dried in a vacuum oven at 80 °C for 1 h to completely remove the solvent.

The PAA film was further imidized on the glass plate by sequential heating at 110 °C, 140 °C, and 170 °C for 30 min at each temperature, followed by 50 min at 195 °C and 220 °C each. Finally, the reaction was completed by heat treatment at 235 °C for 2 h.

The thickness of the heat-treated film was approximately 67–70 µm. Table 1 summarizes the detailed heat treatment conditions for obtaining PAA, PAA hybrid, and PI hybrid film. The structural formula for the synthetic route is shown in Figure 1.

### 2.3. Characterization

The ^13^C chemical shifts for the BAS and BAS-OH were obtained using ^13^C cross-polarization/magic angle spinning nuclear magnetic resonance (CP/MAS NMR) at Larmor frequencies of ω_0_/2π = 100.61 MHz using Bruker AVANCE II+ 400 MHz NMR spectrometers (Berlin, Germany) at the Korea Basic Science Institute, Western Seoul Center. The chemical shifts were referenced to tetramethylsilane (TMS). Powdered samples were placed in a 4-mm magic angle spinning (MAS) probe and the MAS rate was set to 10 kHz to minimize the spinning sideband overlap.

Differential scanning calorimeter (DSC) was measured using a NETZSCH DSC 200F3 (Berlin, Germany) instrument, and thermogravimetric analysis (Auto TGA 1000) was performed on a TA instrument (New Castle, DE, USA) at a heating rate of 20 °C/min. Both DSC and TGA were operated under nitrogen conditions. The thermal expansion coefficient (CTE) of the film was obtained using a macroexpansion probe (TMA-2940) with a 0.1-N expansion force at a heating rate of 5 °C/min and a temperature range from 50 °C to 150 °C.

Wide angle X-ray diffraction (XRD) values were obtained on a Rigaku (Tokyo, Japan) (D/Max-IIIB) X-ray diffractometer using Ni-filtered Cu-Kα radiation at room temperature. The scanning speed was 2°/min in the 2θ range = 2–12°. Transmission electron microscopy (TEM) photographs of ultra-thin sections of PI hybrid films containing various concentrations of Cloisite 30B were obtained using a Leo 912 OMEGA TEM (Tokyo, Japan) at an accelerating voltage of 120 kV. The yellow index (YI) value of the polymer film was measured using a Minolta spectrophotometer (Model CM-3500d, Tokyo, Japan), and UV-Vis spectral results were obtained using Shimadzu UV-3600 (Tokyo, Japan).

The tensile properties of the films were measured using an Instron (Seoul, Korea) mechanical tester (Model 5564) with a crosshead speed of 20 mm/min at room temperature. Experimental errors of the obtained tensile strength and elastic modulus were within ± 1 MPa and ± 0.05 GPa, respectively. Average values of at least 20 samples were finally used for analysis.

## 3. Results and Discussion

### 3.1. FT-IR and NMR Spectra

Figure 2 shows the IR results of monomers and polymers. In the BAS polymer, C=O peaks were observed at 1714 and 1762 cm^−1^. C-N-C peaks indicating the imidization of the polymers were observed at 1377 cm^−1^. The spectrum of the BAS-OH monomer, the primary amine -NH2 was observed at 3374 cm^−1^, and the -OH peak was also observed at 3225 cm^−1^. In addition, C=C aromatic stretch. Peaks appeared at 1746, 1667, and, 1536 cm^−1^, respectively.

In the BAS-OH polymer, the peak of O-H stretching was observed at 3324 cm^−1^. The -OH peaks in BAS-OH were smaller than typical hydroxyl peaks. The O-H stretching absorption of the hydroxyl group is sensitive to hydrogen bonding. Molecules with hydrogen donors and acceptors capable of intramolecular hydrogen bonding in the PI main chain show a broad O-H stretching absorption in the range from 3000 to 3500 cm^−1^. The spectrum of the BAS-OH polymer in Figure 2 shows the hydrogen-bonded peak between -OH in the phenols and the nitrogen in the adjacent imides [27]. In addition, similar to the BAS polymer, the C-N-C peak was observed at 1371 cm^−1^ [28]. These results show that both PIs exhibited a completed imidization reaction.

Structural analyses of the BAS and BAS-OH polymers were carried out by solid state ^13^C CP/MAS NMR [26]. The ^13^C chemical shifts of the BAS polymer were obtained at room temperature. The chemical shift for carbon in 4, 4′-hexafluoroisopropylidene (HFP) was present at 65.38 ppm, as shown in Figure 3a. Here, the peaks of 126.38, 131.73, and 138.51 ppm were attributable to the carbon in aromatic ring and CF_3_, and the chemical shift of C=O was 165.23 ppm. The resonance peak for the carbon in 4,4′-hexafluoroisopropylidene (HFP) had a smaller intensity. The spinning sidebands are marked with an asterisk. The chemical shifts for all carbons were consistent with the structure shown in Figure 3a.

On the other hand, the chemical shift for carbon in 4,4′-hexafluoroisopropylidene (HFP) in the BAS-OH polymer was present at 64.99 ppm, as shown in Figure 3b. The signals at 119.21, 124.95, 132.44, and 137.41 ppm were attributable to the aromatic ring and -CF_3_. In addition, the ^13^C chemical shift at 157.74 and 165.94 ppm corresponding to C-OH and C = O, respectively, was consistent with the structure shown in Figure 3b.

### 3.2. XRD

Figure 4 shows the XRD patterns of pure organoclay, PI, and PI hybrid films with organoclay concentrations ranging from 0.25 wt% to 1.00 wt%. In the case of the BAS/PI hybrid (Figure 4), the XRD peak of Cloisite 30 B was observed at 2θ = 4.76°, which represents an interlayer distance of 18.56 Å. For all PI hybrids containing 0.25 wt% to 1.00 wt% of Cloisite 30 B, no clay peaks were observed in the XRD curves. Similarly, in the case of BAS-OH/PI (Figure 4), clay peaks were not observed at all in the XRD curves for all PI hybrids containing 0.25–0.75 wt% of Cloisite 30 B for PI. However, when 1.00 wt% Cloisite 30 B was used, a very weak peak was observed at 2θ = 6.62° (d = 13.35 Å). This peak is due to the agglomeration of clay because of the use of excess organoclay.

In both the hybrids, we found that clay was very uniformly distributed in the PI matrix regardless of the clay concentration. This denotes that the clay layers in the hybrid material are homogeneously dispersed in the PI matrix [29,30].

XRD is a useful technique for measuring d-spacing in dispersed clay layers of hybrids. In particular, studies using small-angle X-ray scattering can provide detailed information on composites. Schneider et al. [31] reported the influence of different interactions between silica surface and rubber chains depending on the fraction of silica.

### 3.3. Morphological Analysis by TEM

Direct evidence regarding the formation of nano-sized composites can be confirmed via TEM of an ultra-microtomed section. TEM was used to more accurately measure the dispersion of clay layers in PI hybrids. Figure 5 shows TEM images of the BAS hybrids with 0.5 wt% and 1.00 wt% Cloisite 30 B. In the case of the 0.50 wt% hybrid, some clay was exfoliated and some was intercalated, as shown in Figure 5, but, for the 1.00 wt% hybrid, clay was more agglomerated rather than dispersed. The result is that clay is not evenly dispersed and some of the clay is agglomerated (see Figure 5). As the content of Cloisite 30 B in the PI increased, clay aggregation also increased. The agglomerated clay exhibited poor cohesion and compatibility with PI, thus weakening the interfacial adhesion between the polymer matrix and the filler [22,32,33]. These factors eventually reduce the thermal properties of the hybrid, as described in the next section.

Hydrophilic clay has good affinity with PI synthesized with BAS-OH monomers, which include an -OH group. Thus, good affinity of clay and PI can result in excellent dispersion. The TEM results of the BAS-OH hybrid are shown in Figure 6. Unlike the BAS hybrid, BAS-OH showed good dispersion in both 0.50 wt% and 1.00 wt% Cloisite 30 B. At 0.50 wt% concentration (see Figure 6), it was confirmed that the clay was uniformly distributed in the PI matrix with a size of less than 10 nm, and this result was similar at 1.00 wt%. Compared with the results of 0.50 wt% hybrid, some clay was agglomerated to a size of less than 20 nm, but most clay was distributed evenly below 20 nm (see Figure 6). When BAS and BAS-OH hybrids with the same clay concentration were compared, the degree of the BAS-OH clay dispersion was better than that of the BAS hybrid. These results suggest that the good hydrophilicity of BAS-OH can be explained by its high affinity for clay.

### 3.4. Thermal Behavior

Table 2 shows the thermal properties of PI hybrids with various clay contents. The *T_g_*s of PI hybrid films containing BAS monomer increased gradually from 227 °C to 245 °C as the clay content increased from 0 wt% to 0.50 wt%. This increase in *T_g_* is caused by the difficulty of movement of the inserted polymer chains in the clay layer, which results in difficulty in segmental motion [34,35]. However, the *T_g_* of the PI hybrid increased when organic clay content was increased to a certain critical concentration but decreased when the critical concentration was crossed. For example, when Cloisite 30 B was increased to 0.75 wt%, the *T_g_* of the PI hybrid decreased to 240 °C, and, when the organoclay content of PI reached 1.00 wt%, the *T_g_* further decreased to 236 °C.

The trends exhibited by PI hybrids containing BAS-OH monomers were similar to those exhibited by PI hybrids containing BAS. When Cloisite 30B content was increased from 0 wt% to 0.50 wt%, *T_g_* increased from 259 °C to 270 °C. However, when the organoclay content was increased to 1.00 wt%, *T_g_* decreased to 257 °C. This reduction in *T_g_* was because of the agglomeration of excess clay above the critical concentration in the PI matrix. Figure 7 shows the DSC thermogram of PI hybrids according to the concentration of Cloisite 30 B. Clay aggregation above the critical concentration has already been demonstrated using TEM (see Figure 5 and Figure 6).

In general, *T_g_* is affected by various factors in addition to the critical concentration of the filler. That is, the *T_g_* of a polymer depends on the structural differences within the chain, chemical interactions such as hydrogen bonding and curing reaction, the chain flow according to the free volume, and the presence of additives [36]. When the *T_g_* values of the PI hybrids—BAS and the BAS-OH— and those of the two monomers were compared, the *T_g_* value of the BAS-OH PI hybrids were found to be higher than those of the BAS PI hybrid, regardless of the concentration of organic clay. In addition, when the *T_g_* values of the hybrids with the same Cloisite 30 B concentrations were compared, the *T_g_* value of BAS-OH was found to be higher than that of BAS. These results show that the -OH groups of the BAS-OH monomers increase the dispersibility and compatibility of the monomers through hydrogen bonding with the -OH groups present in the clay, thereby increasing the *T_g_* values of the PI hybrids. Similar results have been obtained in previous studies [37,38].

Table 2 shows the initial decomposition temperatures (*T*_D_^i^) of the PI hybrid film containing BAS monomer according to the concentration of the organoclay. As Cloisite 30 B content increased from 0 wt% to 0.50 wt%, the *T*_D_^i^ value of the PI hybrids increased gradually from 456 °C to 533 °C (see Table 2). This suggests that the clay in the polymer chains acts as an insulator and a barrier against volatile products generated during heating, thereby increasing the initial decomposition temperature. This increase in thermal stability is caused by the high thermal stability of the clay itself and the interaction between clay particles and the polymer matrix [24,39]. However, when 1.00 wt% organoclay was added to the PI hybrid (521 °C), the *T*_D_^i^ value was lower than that of the 0.50 wt% hybrid by 12 °C. As has already been explained, the decrease in the *T*_D_^i^ value was because of the aggregation of the excess clay used. Figure 8 shows the TGA thermogram of the PI hybrids having various concentrations of organoclay. The same tendency was also observed for PI hybrids containing BAS-OH monomers. That is, *T*_D_^i^ showed a maximum value of 330 °C for the 0.50 wt% hybrid B, but the *T*_D_^i^ value for the 1.00 wt% hybrid was reduced to 316 °C (see Table 2). The values of the weight residue at 600 °C (*wt_R_^600^*) were almost the same regardless of the clay loading, as shown in Table 2 and Figure 8. In the case of PI containing BAS without organic clay, the weight residue was 55%. When Cloisite 30 B content increased from 0.25 wt% to 1.00 wt%, *wt_R_^600^* was 79–84%. However, when the content of Cloisite was 0–1.00 wt%, the value of *wt_R_^600^* was almost constant at 60–62% for PI containing BAS-OH.

We compared the *T*_D_^i^ and *wt_R_^600^* values of the two different PIs containing BAS and BAS-OH monomers. The BAS hybrid showed an overall better thermal stability than the BAS-OH PI hybrid regardless of the concentration of Cloisite 30 B. These values conflicted with the *T_g_* results. This result can be explained by the weak thermal stability of the -OH group present in the main PI chain containing BAS-OH. In the curve of Figure 8, the first weight loss at around 300 °C was thought to be due to the thermal decomposition of the -OH group of the BAS-OH monomer [25].

The CTE of the BAS PI hybrids decreased to the minimum at the critical content of 0.50% clay but then increased when the concentration of organoclay increased to 1.00% by weight. For example, the CTE of both PI hybrids decreased from 47.21 ppm/°C to 38.48 ppm/°C when the clay concentration increased from 0 wt% to 0.50 wt% but increased to 45.92 ppm/°C when the clay concentration increased to 1.00 wt%. The CTE values of the PI hybrids according to different organoclay concentrations are summarized in Table 2. The same tendency was also observed in the BAS-OH/PI hybrids. The CTE value of the was decreased from 53.17 ppm/°C to 49.95 ppm/°C when up to 0.50 wt% of clay was added but increased to 61.17 ppm/°C when clay concentration increased to 1.00 wt%.

These observations regarding CTE values depend on the orientation of the straight plate-like clay itself, the shape of the PI polymer embedded in the clay layer, and the cohesive strength of clay and PI [40]. When heated, the PI molecules oriented in plane tended to expand in a direction perpendicular to the original direction and thus expanded mainly in the out-of-plane direction [41]. However, because the clay layer in the hybrid is much straighter and stronger than the PI molecule, it is not easily deformed or expanded like a PI molecule. As a result, the existing clay layer in the hybrid can very effectively inhibit the thermal expansion of the PI molecules in the out-of-plane direction [42,43]. However, in our system, the CTE values increased when the concentration of organoclay exceeded 0.50 wt%. This result may be because of clay aggregation, as has been already explained. These results are supported by the *T_g_* and *T*_D_^i^ results previously described (Table 2). The CTE results of the BAS hybrids were better than those of the BAS-OH hybrids at the same organoclay concentration, as seen from the thermal stability analysis. These results are also due to the -OH groups with low thermal stability.

### 3.5. Mechanical Tensile Properties

The ultimate tensile strength, initial tensile modulus, and elongation percentage at break of the PI hybrid films with various organoclay contents were measured using a universal tensile machine (UTM), and the obtained results are summarized in Table 3. The mechanical properties of the PI hybrid film were improved when the clay concentration was below the critical concentration but deteriorated when the critical clay content was exceeded, as already confirmed from the thermal properties according to the critical concentration of clay. For example, when the Cloisite 30 B content increased from 0 wt% to 0.5 wt%, the tensile strength increased from 53 MPa to 103 MPa for BAS hybrids and from 59 MPa to 112 MPa for BAS-OH hybrids. These results show a remarkable increase of 194% and 190%, respectively, using a small amount of organoclay (0.50 wt%). However, when the concentration of organoclay became 1.00 wt%, the tensile strength decreased to 69 MPa for the BAS PI hybrid and 82 MPa for the BAS-OH PI hybrid, as shown in Table 3. Similar results have been reported for other polymer hybrids. Yano et al. [44,45] reported that the mechanical properties of cellulose/silica composites were improved up to a certain concentration of silica, but the tensile strengths of the hybrids reduced at higher filler concentrations [46]. These tendencies occur mainly because the filler particles existing above the critical concentration are not dispersed evenly and are aggregated with each other [20]. This phenomenon has already been confirmed from the TEM photograph in Figure 5 and Figure 6.

However, the initial modulus increased steadily with increasing organoclay concentration. As the concentration of Cloisite 30 B increased from 0 wt% to 1.00 wt%, the tensile modulus increased steadily from 2.24 GPa to 3.54 GPa for the BAS PI hybrid and from 2.80 GPa to 6.16 GPa for the BAS-OH PI hybrid. In contrast to the results of ultimate tensile strength, this improvement in the initial tensile modulus of the two hybrids was due to the high aspect ratio and orientation of the clay layer as well as the resistance of the clay itself to external forces [47,48].

We compared the mechanical properties of the two PI hybrids and those of the BAS and BAS-OH monomers. The mechanical properties of BAS-OH hybrids were better than those of the BAS hybrids regardless of the concentration of organoclay. This tendency was in contrast to the tendency of thermal stability and CTE. This result is attributed to the formation of a hybrid film with stronger bonds that can withstand external tensile forces because the hydroxyl groups between PI and organoclay form hydrogen bonds with each other. Figure 9 shows the change in the mechanical properties of the two PI hybrid films according to the various organoclay contents.

The elongation at break of the two hybrid films was almost constant regardless of the organoclay concentration. That is, when the organoclay content increased up to 1.00 wt%, the elongation at break varied from 2% to 4% (see Table 3). This result is characteristic of hybrid materials reinforced with a rigid and hard inorganic material.

### 3.6. Optical Transparency

The color intensity can be determined by measuring the cut-off wavelength (λ_0_) using UV-Vis. absorption spectra, as shown in Figure 10. The results of the absorption spectrum of the BAS hybrid are listed in Table 4. As the Cloisite 30 B concentration increased from 0 wt% to 1.00 wt% in the BAS and BAS-OH hybrids, the value of λ_0_ increased steadily from 330 nm to 346 nm and from 344 nm to 350 nm, respectively. PI hybrids showed a small λ_0_ value because they have a -CF_3_ substituent and a kinked monomer structure that can reduce the interactions between the CT-complexes and the intermolecular interaction in the polymer main chain [15,16,49].

The transmittance at 500 nm gradually decreased as the organoclay concentration increased to 1.00 wt%. For example, for BAS and BAS-OH hybrids, the transmittance at 500 nm decreased from 87% to 82% and from 84% to 75%, respectively. The color intensities of pure PI and PI hybrid films containing various concentrations of Cloisite 30 B are summarized in Table 4. Color intensity can be described as an index indicating the value of yellow index (YI). The YI of the pure PI film was 2, but those of the BAS hybrid films slightly increased as the organoclay content increased because of the agglomeration of the clay particles [43]. However, the difference was not large, so it showed a value of 2–4. In the case of BAS-OH hybrids, the YI of the hybrids increased as Cloisite 30 B content from 0 to 0.50 wt% and then remained constant at 14. However, when the organoclay content increased to 1.00 wt%, the YI increased significantly to 20.

The reason that the optical transparency is gradually lowered as the clay concentration increases is that clay agglomeration affects optical properties. Evidence for clay agglomeration has already been demonstrated using XRD and TEM (see Figure 4, Figure 5 and Figure 6). The optical transparency of BAS-OH hybrids is lower than that of the BAS hybrids because of the strong intermolecular bonding due to the hydrogen bonding of -OH present in BAS-OH and the consequent increase in the CT-complex, as described above.

As shown in Figure 11, the BAS hybrids with a 0 wt% to 1.00 wt% concentration of Cloisite 30 B were almost colorless and transparent regardless of the organoclay concentration. This transparency and colorlessness suggest that, even when 1.00 wt% of organoclay is added, the phase region of the hybrid film is significantly smaller than the visible light wavelength (i.e., 400 nm to 800 nm). In contrast, in the case of BAS-OH hybrids, a yellowish film was obtained as a whole, and this color became darker as the concentration of Cloisite 30 B increased to 1.00 wt% (see Figure 12). However, both hybrids exhibited excellent transparency, so that there was no problem in reading the text through the film for all organoclay concentrations.

## 4. Conclusions

Two PI hybrid films were fabricated from dianhydrides (6FDA), diamines (BAS and BAS-OH), and various concentrations of organoclay by solution intercalation. The PI hybrid films containing -CF_3_ substituents and bent monomer structures showed higher optical transparency than conventional linear non-fluorinated PI films. In particular, the -CF_3_ groups present in the main chain were effective in strongly attracting electrons to reduce the CT-complex between polymer chains through steric hindrance and the induction effect.

In this study, the PI hybrids synthesized using BAS and BAS-OH as diamines were compared with each other according to various Cloisite 30 B concentrations. In terms of *T_g_* and mechanical properties, BAS-OH hybrids showed better performance than BAS hybrids. However, BAS hybrids were superior to BAS-OH hybrids in terms of thermal stability, CTE, and optical transparency over the range of organoclay concentrations. The thermal properties and ultimate strengths of the two PI hybrid films were highest when the organic clay was 0.5 wt%.

In addition, these PI hybrid films are expected to be useful as high-performance polymer materials because of their excellent thermo-mechanical properties and optical transparencies as compared with general engineering plastics.

## Figures and Tables

**Figure 1 polymers-12-00135-f001:**
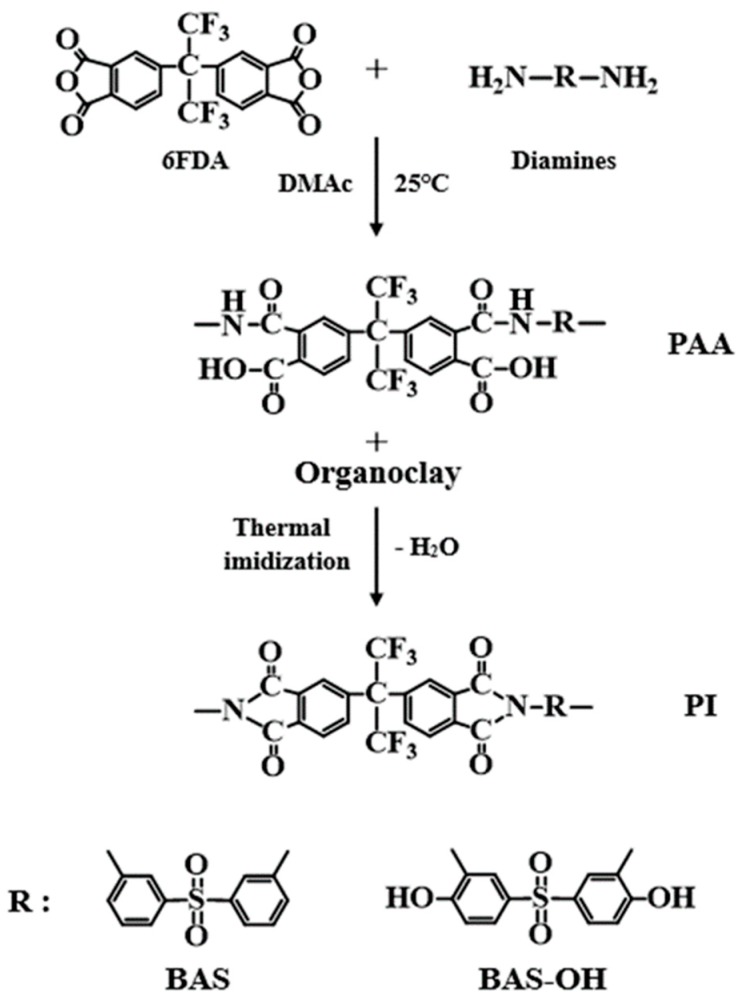
Synthetic route of PI hybrids with various diamines.

**Figure 2 polymers-12-00135-f002:**
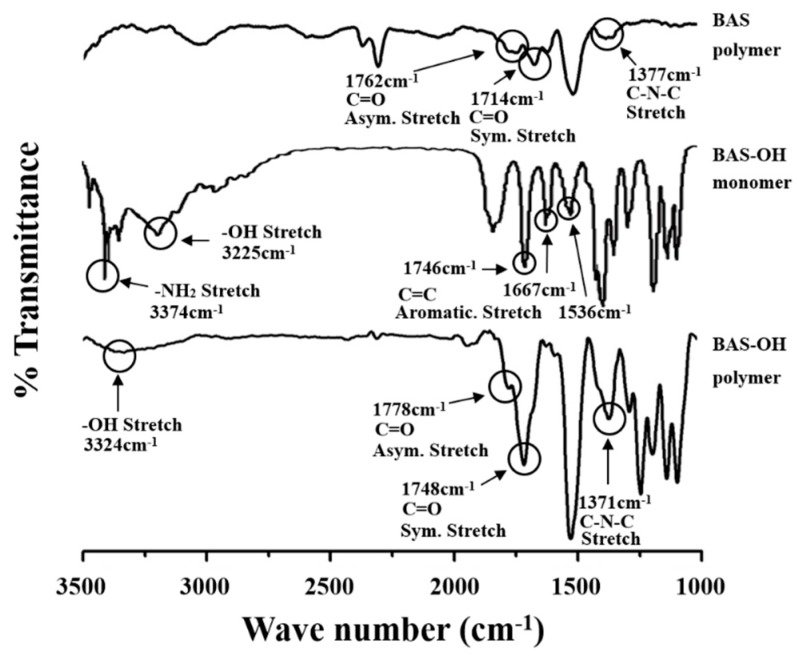
FT-IR spectra of monomer and PIs.

**Figure 3 polymers-12-00135-f003:**
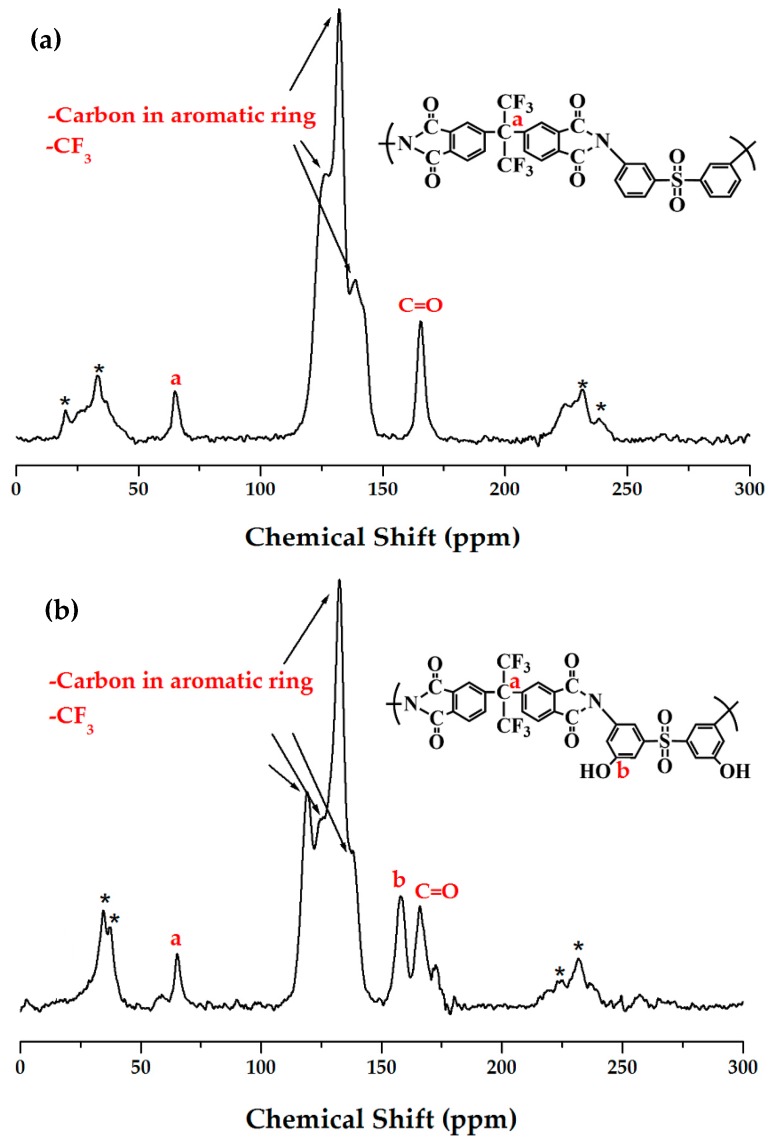
^13^C-NMR chemical shifts of (**a**) BAS and (**b**) BAS-OH polymers at room temperature. The spinning sidebands are marked with asterisks.

**Figure 4 polymers-12-00135-f004:**
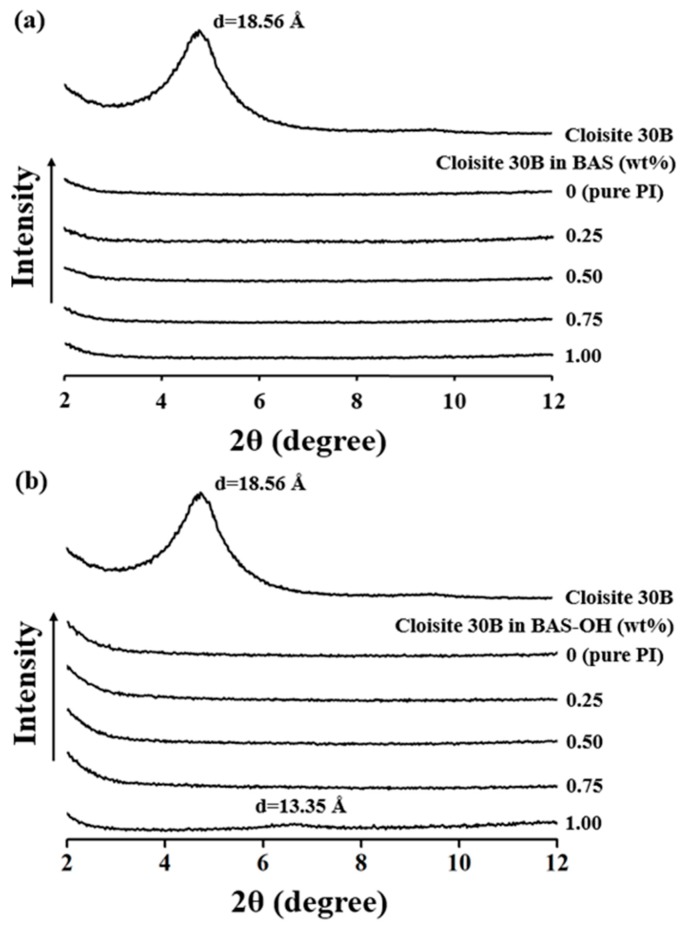
Wide angle X-ray patterns of PI hybrid films with various organoclay. contents: (**a**) BAS and (**b**) BAS-OH.

**Figure 5 polymers-12-00135-f005:**
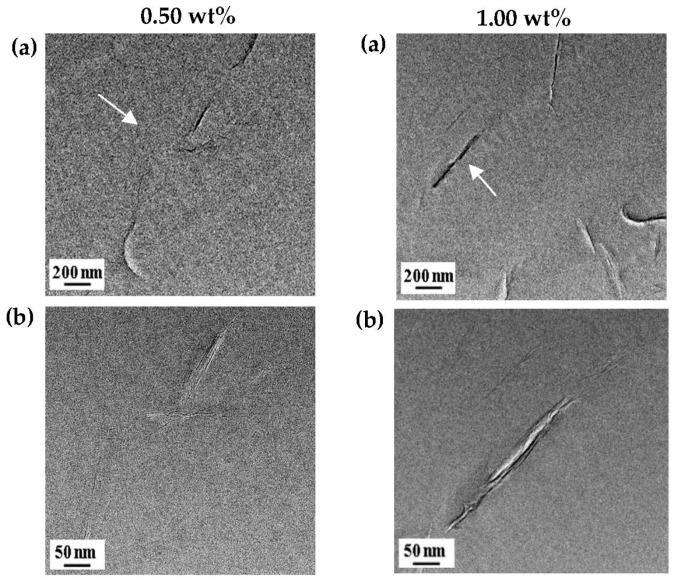
TEM micrographs of BAS hybrid films containing 0.5 wt% and 1.00 wt%. Cloisite 30 B with increasing magnification levels from (**a**) to (**b**).

**Figure 6 polymers-12-00135-f006:**
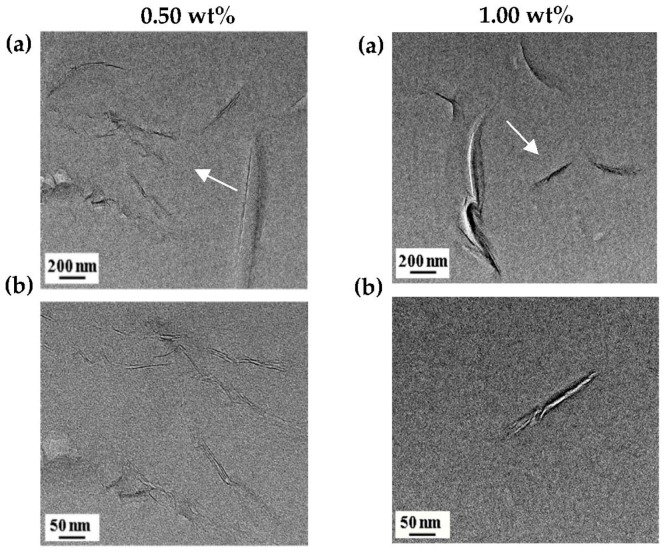
TEM micrographs of BAS-OH hybrid films containing 0.5 wt% and 1.00 wt%. Cloisite 30 B with increasing magnification levels from (**a**) to (**b**).

**Figure 7 polymers-12-00135-f007:**
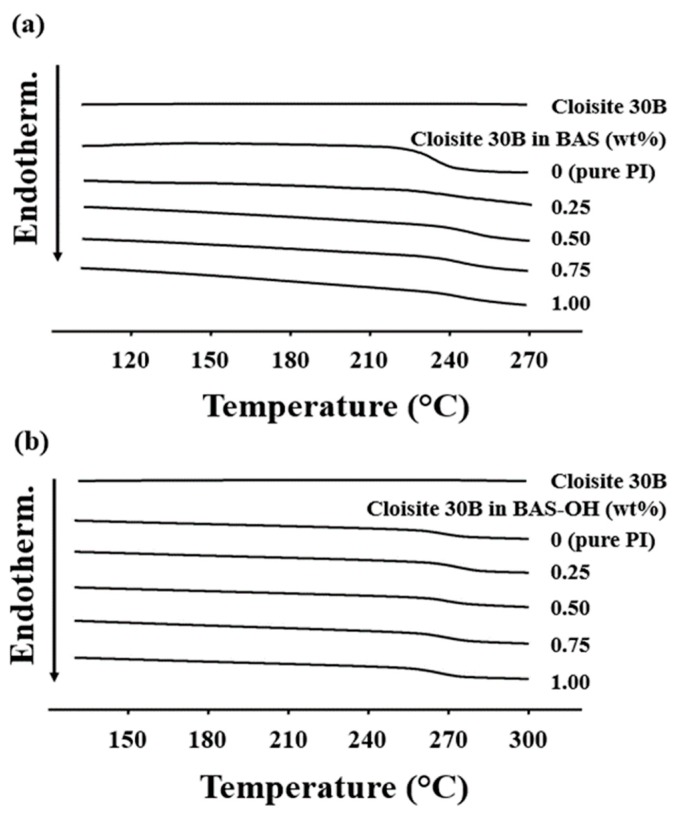
DSC thermograms of PI hybrid films with various organoclay. contents: (**a**) BAS and (**b**) BAS-OH.

**Figure 8 polymers-12-00135-f008:**
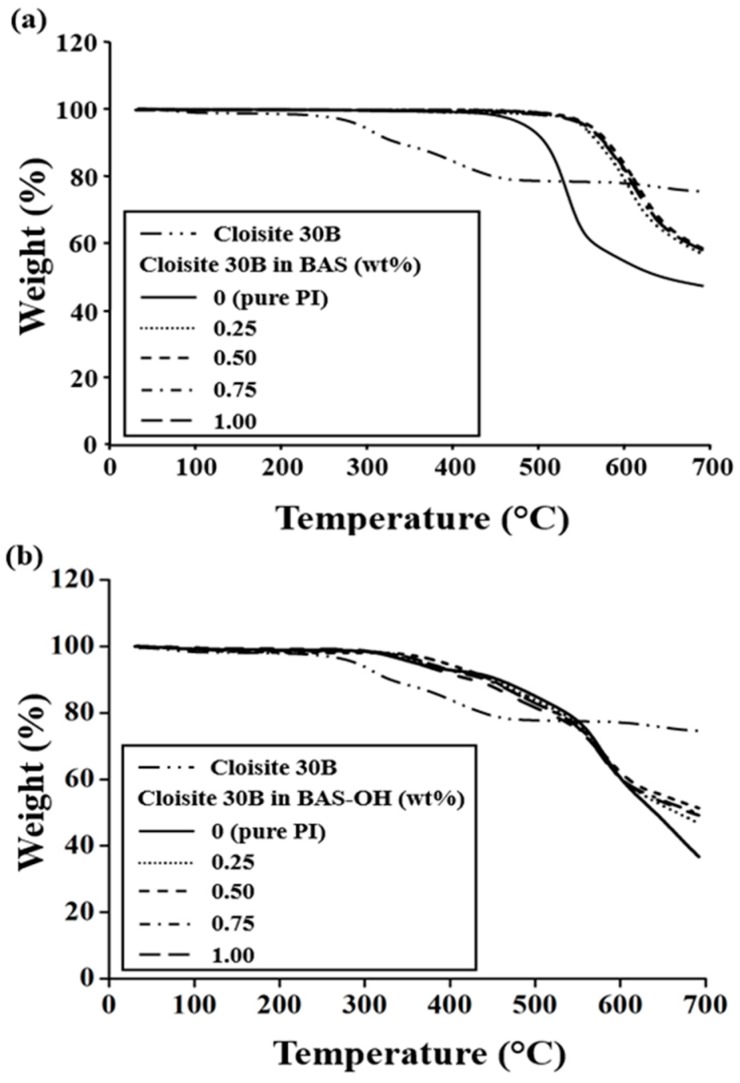
TGA thermograms of PI hybrid films with various organoclay contents: (**a**) BAS and (**b**) BAS-OH.

**Figure 9 polymers-12-00135-f009:**
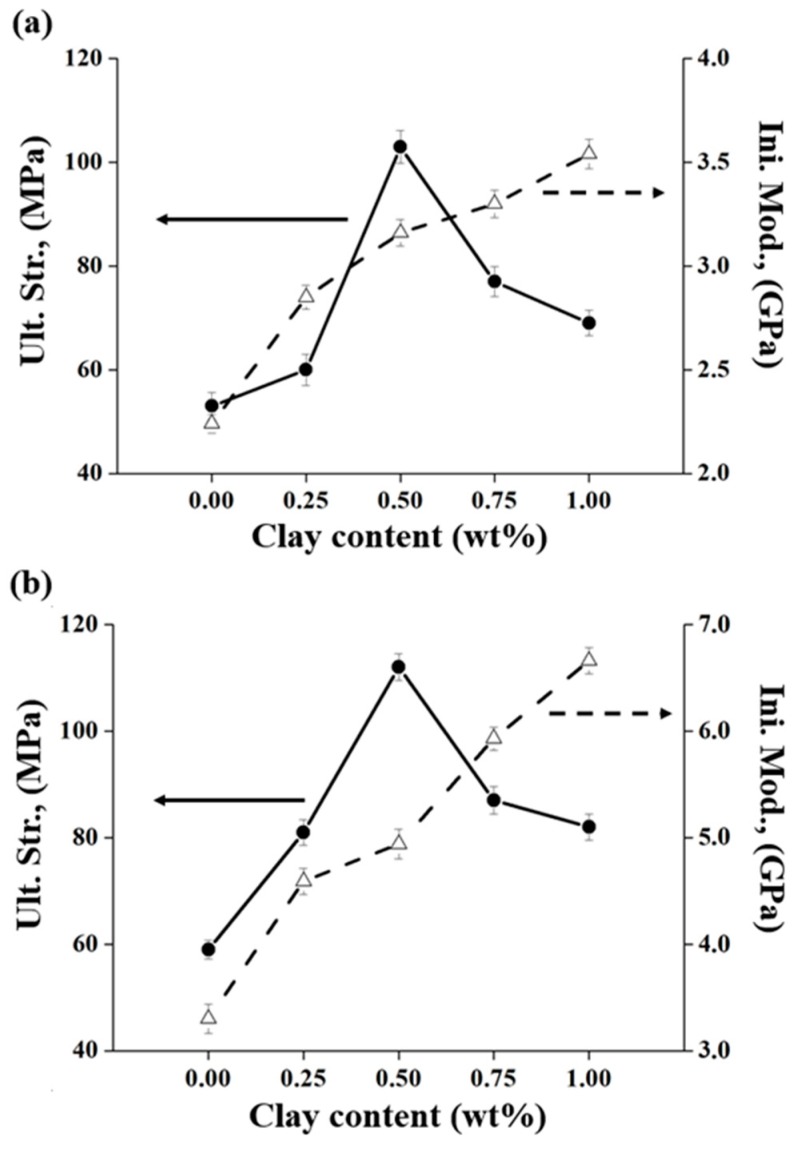
Tensile properties of PI hybrid films with various organoclay contents: (**a**) BAS and (**b**) BAS-OH.

**Figure 10 polymers-12-00135-f010:**
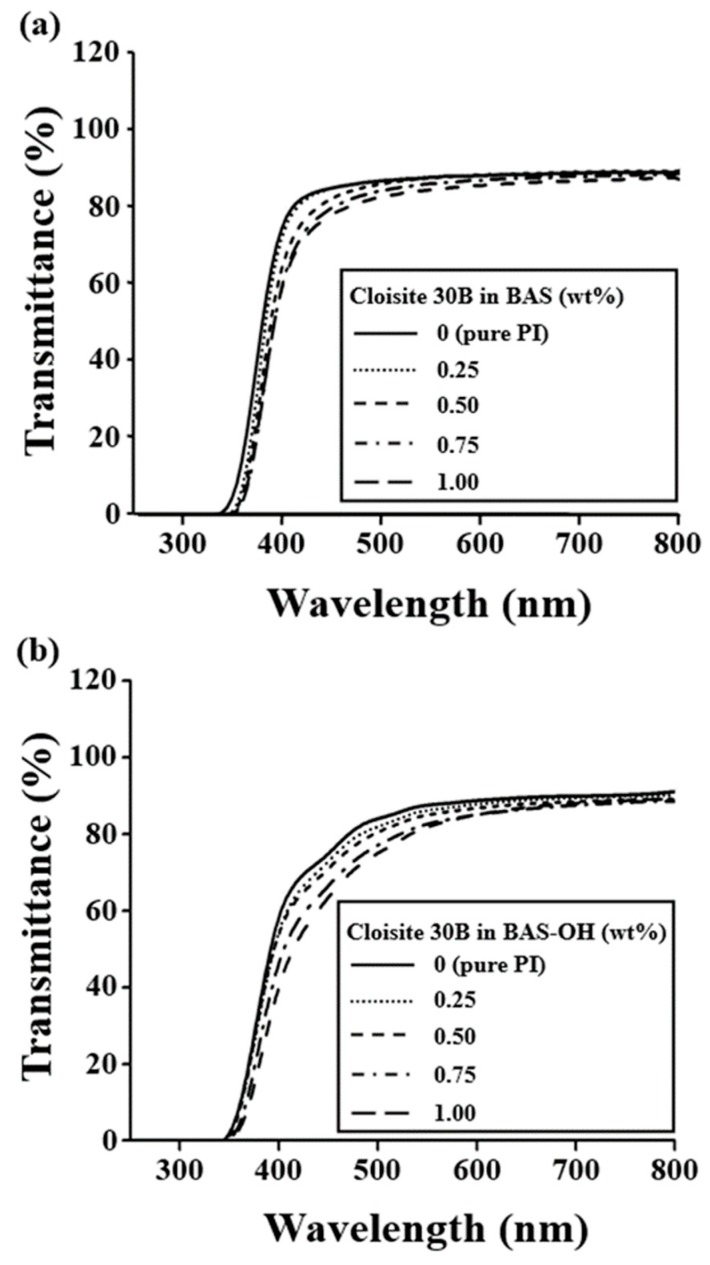
UV-Vis transmittance of the PI hybrid films with various organoclay contents: (**a**) BAS and (**b**) BAS-OH.

**Figure 11 polymers-12-00135-f011:**
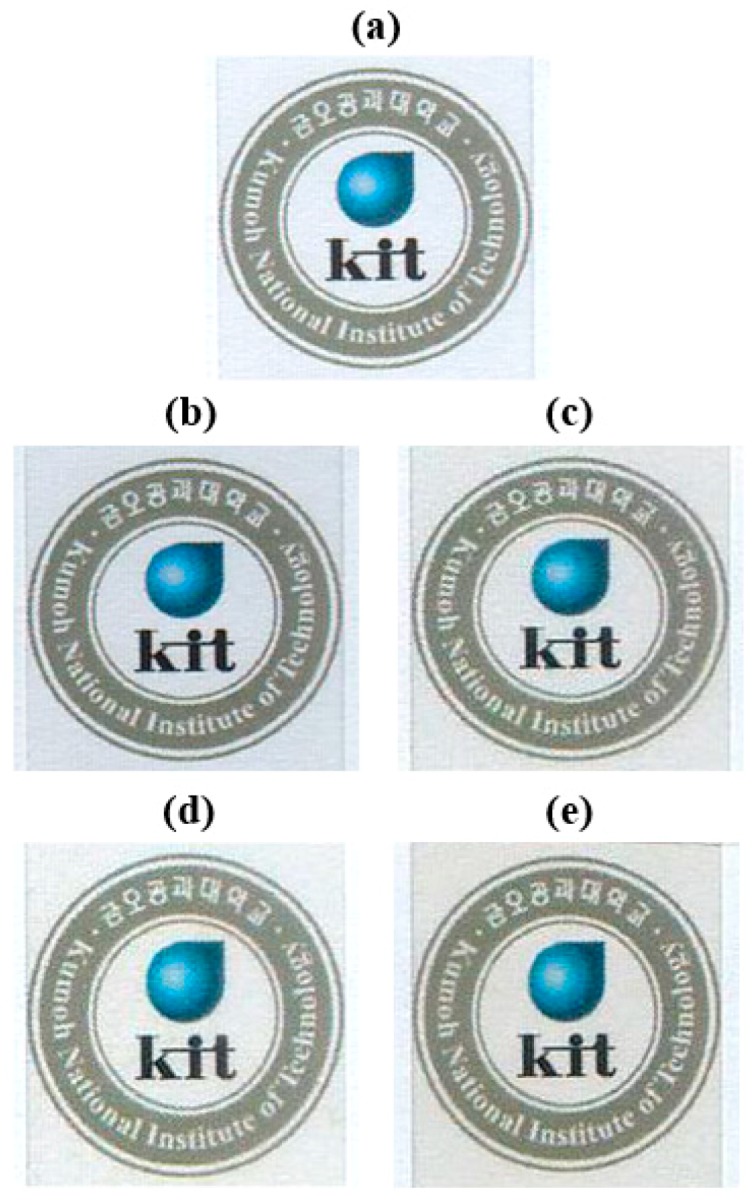
Photographs of PI hybrid films with BAS monomer containing. (**a**) 0 (pure PI), (**b**) 0.25, (**c**) 0.50, (**d**) 0.75, and (**e**) 1.00 wt% of Cloisite 30 B.

**Figure 12 polymers-12-00135-f012:**
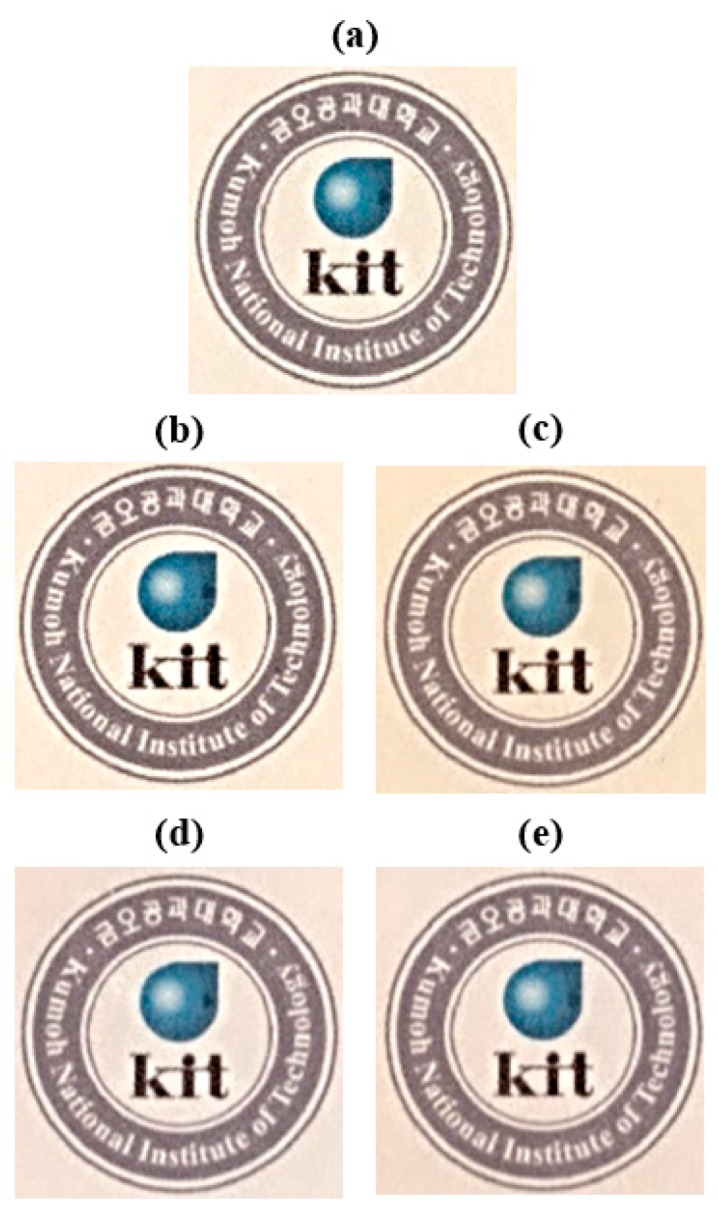
Photographs of PI hybrid films with BAS-OH monomer containing. (**a**) 0 (pure PI), (**b**) 0.25, (**c**) 0.50, (**d**) 0.75, and (**e**) 1.00 wt% of Cloisite 30 B.

**Table 1 polymers-12-00135-t001:** Heat treatment conditions of PI hybrid films.

Samples	Temp. (°C)/Time (h)/Pressure (Torr)
PAA	0/1/760 → 25/14/760
PAA hybrid	25/6/760
PI hybrid	50/2/1 → 80/1/1 → 110/0.5/1 → 140/0.5/1 → 170/0.5/1 → 195/0.8/1 → 220/0.8/1 → 235/2/1

**Table 2 polymers-12-00135-t002:** Thermal properties of PI hybrid films with various organoclay contents.

Cloisite 30B (wt%)	BAS				BAS-OH			
	*T_g_* (°C)	*T*_D_^i^^a^ (°C)	*wt_R_^600^*^b^ (%)	CTE ^c^ (ppm/°C)	*T_g_* (°C)	*T*_D_^i^ (°C)	*wt_R_^600^* (%)	CTE (ppm/°C)
0 (pure PI)	227	456	55	47.21	259	313	60	53.17
0.25	231	526	79	41.64	263	321	61	49.95
0.50	245	533	84	38.48	270	330	61	48.61
0.75	240	530	82	42.37	261	324	60	54.33
1.00	236	521	83	45.92	257	316	62	61.17

^a^ At a 2% initial weight-loss temperature. ^b^ Weight percent of residue at 600 °C. ^c^ Coefficient of thermal expansion for 2nd heating is 50–150 °C.

**Table 3 polymers-12-00135-t003:** Tensile properties of PI hybrid films with various organoclay contents.

Cloisite 30B (wt%)	BAS			BAS-OH		
	Ult. Str. (MPa)	Ini. Mod. (GPa)	E.B. ^a^ (%)	Ult. Str. (MPa)	Ini. Mod. (GPa)	E.B. (%)
0 (pure PI)	53	2.24	2	59	2.80	2
0.25	60	2.85	2	81	4.09	3
0.50	103	3.16	4	112	4.44	3
0.75	77	3.30	3	87	5.43	2
1.00	69	3.54	2	82	6.16	2

^a^ Elongation percentage at break.

**Table 4 polymers-12-00135-t004:** Optical transparencies of PI hybrid films with various organoclay contents.

Cloisite 30B (wt%)	BAS				BAS-OH			
	T ^a^ (μm)	λ_0_ (nm)	500 nm^trans^ (%)	YI ^b^	T (μm)	λ_0_ (nm)	500 nm^trans^ (%)	YI
0 (pure PI)	67	330	87	2	70	344	84	14
0.25	68	334	87	2	69	344	82	14
0.50	68	336	86	3	68	345	81	14
0.75	67	340	84	3	69	349	77	16
1.00	69	346	82	4	69	350	75	20

^a^ Film thickness. ^b^ Yellow index.

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
