# Peer review of "Transparent Polyimide/Organoclay Nanocomposite Films Containing Different Diamine Monomers"

_polymers, 2020, doi:10.3390/polym12010135_

Round 1

Reviewer 1 Report

Polymers668743

This manuscript describes the preparation of PI-Cloisite 30B hybrid films and their properties. I think the authors must perform major revision on their paper as follows before the publication.

After reading this paper, I came to know that the authors’ goal of this project was the preparation of PI hybrid film for optical application. I recommend the authors to add the targeted properties with detail values (like transparency, Tg, YI, and so on) for their polymer hybrid films in Introduction part to clear the goal of this study. Are the polymers described in this paper new ones? If reported ones, the references must be clearly described. Or, full characterization data including IR, NMR, and elemental analysis for all the polymers should be added in the experimental parts. For the preparation of PAA, the authors described that the solution was vigorously stirred. Is this true? Usually, for PAA synthesis, the stirring should be performed in proper speed because of the high viscosity of the polymerization solution. In line 83, 0.50wt% seems incorrect. In 15.5wt PAA solution in DMAc 13.0g, PAA exists 2.015g, thus the weight fraction of Cloisite 30B should be 3.08wt%. Is this calculation OK? Please check it again. In line 85, “The solution was also performed.” Sounds strange. Please revise the sentence. How did you perform imidization? The atmospheric conditions should be added in line 91. Why did you choose metha substituted diamine monomers? With this combination, high molecular weight polymer cannot be expected. The inherent viscosity data of the polymers (PAA) must be added in the text. In line 127, the authors describe the existence of hydrogen bonding, but the detailed wavelength was not in the text. Which bondings do the absorption come from at 2500 to 2900cm-1 in Figure 2? Why hydrogen bonding can be observed only in polymer film? There is no description on BAS-OH monomer in IR spectrum. Figure 4 and 5 do not make sense for readers. What do the arrows indicate in the pictures? I can see several rod-like objects within the film. What are they? Furthermore, I cannot find differences between Figure 4 and 5. Cloisite 30B should have alkylammonium ligands within the structure, and they should degrade under TGA analysis. Where can we observe? The low TDi of BAS-OH polymers may be attributed from the ligands. Is it right? The Tg data in Table 2 shows similar effect of Cloisite 30B on PAA polymers (about 10-15oC Tg increase compared with the base PAA). I cannot find any advantage of BAS-OH moiety. The data given in Table 3 shows poor mechanical strength of the authors’ bare PI films compared with the ordinal PI films. The authors should clearly mention the reason why they selected this PI samples for hybridization. Film thickness should be added. Table 4 should be revised.

Author Response

Author Response:

Dear Editor

This is my response to your comments regarding our paper “Transparent Polyimide/ Organoclay Nanocomposite Films Containing Different Diamine Monomers (polymers-668743) in polymers.

Thank you very much for the referee's comments. I have carefully revised the manuscript following the comments of the referee.

Response to Reviewer-1 comments:

Point-1: NMR characterization data for the samples.

Answer-1: As the reviewer pointed out, the results for solid 13C NMR are shown graphically in Figure 3. Also, the NMR results are explained in section 3.2..

See page 5, line 157: “Structural analyses of the BAS and BAS-OH polymers were carried out by solid state 13C CP/MAS NMR [26]. The 13C chemical shifts of the BAS polymer were obtained at room temperature. The chemical shift for carbon in 4, 4’-hexafluoroisopropylidene (HFP) was present at 65.38 ppm, as shown in Fig. 3(a). Here, the peaks of 126.38, 131.73, and 138.51 ppm were attributable to the carbon in aromatic ring and CF3, and the chemical shift of C=O was 165.23 ppm. The resonance peak for the carbon in 4, 4’-hexafluoroisopropylidene (HFP) had a smaller intensity. The spinning sidebands are marked with an asterisk. The chemical shifts for all carbons were consistent with the structure shown in Fig. 3(a).

On the other hand, the chemical shift for carbon in 4, 4’-hexafluoroisopropylidene (HFP) in the BAS-OH polymer was present at 64.99 ppm, as shown in Fig. 3(b). The signals at 119.21, 124.95, 132.44, and 137.41 ppm were attributable to the aromatic ring and -CF3. In addition, the 13C chemical shift at 157.74 and 165.94 ppm corresponded to C-OH and C=O, respectively, was consistent with the structure shown in Fig. 3(b).

Point-2: Stirring speed when PAA synthesis.

Answer-2: See page 2, line 89: “This solution was stirred at a moderate speed at 25 °C for 14 h to synthesize a PAA solution having a solid content of 15.5 wt%.”  

Point-3. Calculation for solid content.

Answer-3: See page 2, line 91: “The total 15.5 wt% PAA solution produced by this method was 77.62 g. When 0.39 g of Cloisite 30B was added to the solution, the total solution weight was 78.01 g.”

Point-4: Revise word in “The solution was also performed”.

Answer-4: See page 3, line 97: “The solution was also washed in an ultrasonic cleaner for 3h,”

Point-5: Reasons for using metha-substituted diamine monomers.

Answer-5: The kinked structure or the structure with substituents should be used to prevent CT-complex and to synthesize colorless transparent polyimide as in the case of BAS polyimide. This is explained in detail on page 2. See page 2, line 48: “To reduce the CT-complex, a strong electron withdrawing group such as fluorine (F) or sulfone (-SO2-) is required that can effectively introduce a bending structure which can interfere with the interaction between the PI main chains, thus aiding in the fabrication of a PI film with high transparency. For example, a -CF3 group, which is a strong electron-withdrawing group, is often used as a substituent, or bent monomer structures are used to prevent CT complexes from being formed in linear structures [16,17].

Point-6: The inherent viscosity of the PAA.

Answer-6: See page 2, line 91: “The inherent viscosities of the synthesized PAAs of BAS and BAS-OH were 1.02 and 0.94, respectively.

Point-7: Detailed description of hydrogen bonding in BAS-OH polymer.

Answer-7: A detailed description has been added for the BAS-OH polymer. See page 5, line 149: “In the BAS-OH polymer, the peak of O-H stretching was observed at 3324 cm-1. The -OH peaks in BAS-OH were smaller than typical hydroxyl peaks. The O-H stretching absorption of the hydroxyl group is sensitive to hydrogen bonding. Molecules with hydrogen donors and acceptors capable of intramolecular hydrogen bonding in the PI main chain show a broad O-H stretching absorption in the range from 3000 to 3500 cm-1. The spectrum of the BAS-OH polymer in Figure 2 shows the hydrogen-bonded peak between -OH in the phenols and the nitrogen in the adjacent imides [27]. In addition, similar to the BAS polymer, the C-N-C peak was observed at 1371 cm-1 [28]. These results show that both PIs exhibited a completed imidization reaction.”

Point-8: There is no description on BAS-OH monomer in IR spectrum.

Answer-8: A detailed description has been added for the BAS-OH monomer. See page 4, line 144: “The spectrum of the BAS-OH monomer, the primary amine -NH2 was observed at 3374 cm-1 and the -OH peak was also observed at 3225 cm-1. Also C=C aromatic stretch. peaks were appeared at 1746, 1667, and 1536 cm-1, respectively.”

Point-9: A detailed description of the TEM picture (Figures 5 and 6)

Answer-9: In Figure caption, arrows and magnifications are explained in detail. See figure captions in Figures 5 and 6.

Point-10: The low TDi of BAS-OH polyimide compare to that of BAS polyimide.

Answer-10: A detailed description has been added for the BAS-OH monomer. See page 10, line 322: “This result can be explained by the weak thermal stability of the -OH group present in the main PI chain containing BAS-OH. In the curve of Figure 8, the first weight loss at around 300 °C was thought to be due to the thermal decomposition of the -OH group of the BAS-OH monomer [25].”

Point-11: Explanation of why the Tg value increases when using organoclay.

Answer-11: As the reviewer pointed out, there was no significant change, but the explanation according to the research results was given on page 9, line 283: “When the Tg values of the PI hybrids—BAS and the BAS-OH— and those of the two monomers were compared, the Tg value of the BAS-OH PI hybrids were found to be higher than those of the BAS PI hybrid, regardless of the concentration of organic clay. In addition, when the Tg values of the hybrids with the same Cloisite 30B concentrations were compared, the Tg value of BAS-OH was found to be higher than that of BAS. These results show that the -OH groups of the BAS-OH monomers increase the dispersibility and compatibility of the monomers through hydrogen bonding with the -OH groups present in the clay, thereby increasing the Tg values of the PI hybrids. Similar results have been obtained in previous studies [37,38].

Point-12: Explain why we chose this system.

Answer-12: This paper is to compare the properties of PI containing -OH groups and those without PI. A full explanation is given on page 11, line 373: “We compared the mechanical       properties of the two PI hybrids and those of the BAS and BAS-OH monomers. The mechanical properties of BAS-OH hybrids were better than those of the BAS hybrids regardless of the concentration of organoclay. This tendency was in contrast to the tendency of thermal stability and CTE. This result is attributed to the formation of a hybrid film with stronger bonds that can withstand external tensile forces because the hydroxyl groups between PI and organoclay form hydrogen bonds with each other.”

Point-13: Film thickness should be added.

Answer-13: As the reviewer pointed out, film thickness be added in Table 4.

Many thanks to the reviewer for detailed comment.

I hope this revision is satisfactory for your further process. 

Reviewer 2 Report

The article titled “Transparent Polyimide Nanocomposite Films Containing Different Diamine Monomers” by Chang et. al reported the synthesis of polyimide/Cloisite 30B nanocomposite films with two different diamine monomers w/ or /o hydroxyl (OH) group and studied the effect of OH group and the amount of Cloisite 30B content on optical, thermal, and mechanical properties of the composite materials. The research is well designed and thoroughly carried out. I therefore recommend to be published after minor revision.

Molecular weight (MW) and dispersity (Ð) of polymer is an important parameter strongly effecting properties of the polymer. In the manuscript, there is no report regarding to MW and Ð of the polyimides synthesized. I would ask the authors to add these values, at least for the two parent polymers with 0 wt% Cloisite 30B. Additionally, IR was used to characterize the composites, it would be more convincing if NMR spectra of the two parent polymers are given. In the introduction (L36-38), it was mentioned that most of the synthetic polyimides are insoluble and infusible and thus have poor processability. However, in the paper the authors did not provide any test or comment with regard to the processability of the polyimide composites reported in this research.   In L 85-86, it was stated that “The solution was also performed in an ultrasonic washing machine for 3h”. However, it is not clear whether it is a sperate experiment or it was done followed by stirring at room temperature for 3h. In L102, it should be “Differential scanning calorimetry (DSC) was measured using NETZSCH DSC 200F3 instrument. L150, I would recommend to delete “ As it already known” and combine this paragraph with the above paragraph (L147-149). L156-157, Figure 4 shows TEM images of the BAS hybrids with 0.5 and 1.0 wt% Cloisite 30B. L 159, but for the 1.00 wt% hydrid L255, Figure 7 shows the TGA thermogram of the PI hybrids having various concentrations of organoclay. L 259, as shown in Table 2 and Figure 7. L274, the first weight loss at around 300 degree L353 Table 4, there are data overlap in the yellow index of BAS-OH. L367, the abbreviation of YI should be added as yellow index (YI). L370, the YI of the hybrids increased as Cloisite 30B content increased from 0 to 0.50 wt% L409, the organic clay was 0.5wt%. In order to more precisely reflect the content of the research, I would suggest to add organoclay in the title as “ Transparent Polyimide/Organoclay Nanocomposite Films Containing Different Diamine Monomers”.

Author Response

Dear Editor

This is my response to your comments regarding our paper “Transparent Polyimide/ Organoclay Nanocomposite Films Containing Different Diamine Monomers (polymers-668743) in polymers.

Thank you very much for the referee's comments. I have carefully revised the manuscript following the comments of the referee.

Response to Reviewer-2 comments:

Point-1: Molecular weight and dispersity of polymers.

Answer-1: Most of the PIs are insoluble and infusible. The PI synthesized in this study is also insoluble and infusible. Therefore, the results of TEM to observe the dispersion of clay in the PI matrix are shown in Figures 5 and 6.

Point-2: NMR characterization data for the samples.

Answer-2: As the reviewer pointed out, the results for solid 13C NMR are shown graphically in Figure 3. Also, the NMR results are explained in section 3.2.. See page 5, line 158: “Structural analyses of the BAS and BAS-OH polymers were carried out by solid state 13C CP/MAS NMR [26]. The 13C chemical shifts of the BAS polymer were obtained at room temperature. The chemical shift for carbon in 4, 4’-hexafluoroisopropylidene (HFP) was present at 65.38 ppm, as shown in Fig. 3(a). Here, the peaks of 126.38, 131.73, and 138.51 ppm were attributable to the carbon in aromatic ring and CF3, and the chemical shift of C=O was 165.23 ppm. The resonance peak for the carbon in 4, 4’-hexafluoroisopropylidene (HFP) had a smaller intensity. The spinning sidebands are marked with an asterisk. The chemical shifts for all carbons were consistent with the structure shown in Fig. 3(a).

On the other hand, the chemical shift for carbon in 4, 4’-hexafluoroisopropylidene (HFP) in the BAS-OH polymer was present at 64.99 ppm, as shown in Fig. 3(b). The signals at 119.21, 124.95, 132.44, and 137.41 ppm were attributable to the aromatic ring and -CF3. In addition, the 13C chemical shift at 157.74 and 165.94 ppm corresponded to C-OH and C=O, respectively, was consistent with the structure shown in Fig. 3(b).”

Point-3. Regard to the processability of the PI.

Answer-3: In order to improve the processability of the PI, an alkyl group or a bent structure is mainly introduced into the PI main chain. See page 1, line 38 for a detailed explanation. “For example, PIs containing trifluoromethyl groups have been synthesized that show a high modulus, low thermal expansion coefficient, and good solubility in conventional organic solvents [9–11]. Another method is to use a copolyimide (Co-PI) using a specific monomer. A Co-PI typically possesses much lower molecular regularity than the corresponding homopolyimide [12,13]. This decreased regularity leads to fewer intermolecular interactions that, in turn, results in new characteristics, such as modified thermo-optical and gas permeation properties, and solubilities. Furthermore, the properties of Co-PIs can be adjusted by varying the ratio of the dianhydride and diamine comonomers.”

Point-4: Revise the sentence in “The solution was also performed”.

Answer-4: See page 3, line 98: “The solution was also washed in an ultrasonic cleaner for 3h,”

Point-5, 6, and 7: We corrected the sentences and words as the reviewer pointed out. Many thanks for reviewer’s detailed comments.

Answer-5, 6, and 7: Corrections are marked in blue color on each page.

Point-8: Changing title.

Answer-8: The title was changed as the reviewer pointed out. See page 1, title: “Transparent Polyimide/Organoclay Nanocomposite Films Containing Different Diamine Monomers”

Many thanks to the reviewer for detailed comments.

I hope this revision is satisfactory for your further process.